# Model-Based Relative Entropy Stochastic Search

**Abbas Abdolmaleki**[1,2,3]**, Rudolf Lioutikov**[4]**, Nuno Lau**[1]**, Luis Paulo Reis**[2,3]**,**
**Jan Peters**[4,6]**, and Gerhard Neumann**[5]

1: IEETA, University of Aveiro, Aveiro, Portugal
2: DSI, University of Minho, Braga, Portugal
3: LIACC, University of Porto, Porto, Portugal
4: IAS, 5: CLAS, TU Darmstadt, Darmstadt, Germany
6: Max Planck Institute for Intelligent Systems, Stuttgart, Germany
{Lioutikov,peters,neumann}@ias.tu-darmstadt.de
{abbas.a, nunolau}@ua.pt, lpreis@dsi.uminho.pt

## Abstract

Stochastic search algorithms are general black-box optimizers. Due to their ease
of use and their generality, they have recently also gained a lot of attention in oper-
ations research, machine learning and policy search. Yet, these algorithms require
a lot of evaluations of the objective, scale poorly with the problem dimension, are
affected by highly noisy objective functions and may converge prematurely. To
alleviate these problems, we introduce a new surrogate-based stochastic search
approach. We learn simple, quadratic surrogate models of the objective function.
As the quality of such a quadratic approximation is limited, we do not greedily ex-
ploit the learned models. The algorithm can be misled by an inaccurate optimum
introduced by the surrogate. Instead, we use information theoretic constraints to
bound the 'distance' between the new and old data distribution while maximizing
the objective function. Additionally the new method is able to sustain the explo-
ration of the search distribution to avoid premature convergence. We compare our
method with state of art black-box optimization methods on standard uni-modal
and multi-modal optimization functions, on simulated planar robot tasks and a
complex robot ball throwing task. The proposed method considerably outper-
forms the existing approaches.

## 1 Introduction

Stochastic search algorithms [1, 2, 3, 4] are black box optimizers of an objective function that is
either unknown or too complex to be modeled explicitly. These algorithms only make weak assump-
tion on the structure of underlying objective function. They only use the objective values and don't
require gradients or higher derivatives of the objective function. Therefore, they are well suited
for black box optimization problems. Stochastic search algorithms typically maintain a stochas-
tic search distribution over parameters of the objective function, which is typically a multivariate
Gaussian distribution [1, 2, 3]. This policy is used to create samples from the objective function.
Subsequently, a new stochastic search distribution is computed by either computing gradient based
updates [2, 4, 5], evolutionary strategies [1], the cross-entropy method [7], path integrals [3, 8], or
information-theoretic policy updates [9]. Information-theoretic policy updates [10, 9, 2] bound the
relative entropy (also called Kullback Leibler or KL divergence) between two subsequent policies.
Using a KL-bound for the update of the search distribution is a common approach in the stochastic
search. However, such information theoretic bounds could so far only be approximately applied
either by using Taylor-expansions of the KL-divergence resulting in natural evolutionary strate-
gies (NES) [2, 11], or sample-based approximations, resulting in the relative entropy policy search

(REPS) [9] algorithm. In this paper, we present a novel stochastic search algorithm which is called MOdel-based Relative-Entropy stochastic search (MORE). For the first time, our algorithm bounds the KL divergence of the new and old search distribution in closed form without approximations. We show that this exact bound performs considerably better than approximated KL bounds.

In order to do so, we locally learn a simple, quadratic surrogate of the objective function. The quadratic surrogate allows us to compute the new search distribution analytically where the KL divergence of the new and old distribution is bounded. Therefore, we only exploit the surrogate model locally which prevents the algorithm to be misled by inaccurate optima introduced by an inaccurate surrogate model.

However, learning quadratic reward models directly in parameter space comes with the burden of quadratically many parameters that need to be estimated. We therefore investigate new methods that rely on dimensionality reduction for learning such surrogate models. In order to avoid over-fitting, we use a supervised Bayesian dimensionality reduction approach. This dimensionality reduction technique avoids over fitting, which makes the algorithm applicable also to high dimensional problems. In addition to solving the search distribution update in closed form, we also upper-bound the entropy of the new search distribution to ensure that exploration is sustained in the search distribution throughout the learning progress, and, hence, premature convergence is avoided. We will show that this method is more effective than commonly used heuristics that also enforce exploration, for example, adding a small diagonal matrix to the estimated covariance matrix [3].

We provide a comparison of stochastic search algorithms on standard objective functions used for benchmarking and in simulated robotics tasks. The results show that MORE considerably outperforms state-of-the-art methods.

## 1.1 Problem Statement

We want to maximize an objective function $R(\boldsymbol{\theta}) : \mathbb{R}^n \rightarrow \mathbb{R}$. The goal is to find one or more parameter vectors $\boldsymbol{\theta} \in \mathbb{R}^n$ which have the highest possible objective value. We maintain a search distribution $\pi(\boldsymbol{\theta})$ over the parameter space $\boldsymbol{\theta}$ of the objective function $R(\boldsymbol{\theta})$. The search distribution $\pi(\boldsymbol{\theta})$ is implemented as a multivariate Gaussian distribution, i.e., $\pi(\boldsymbol{\theta}) = \mathcal{N}(\boldsymbol{\theta}|\boldsymbol{\mu}, \boldsymbol{\Sigma})$. In each iteration, the search distribution $\pi(\boldsymbol{\theta})$ is used to create samples $\boldsymbol{\theta}^{[k]}$ of the parameter vector $\boldsymbol{\theta}$. Subsequently, the (possibly noisy) evaluation $R^{[k]}$ of $\boldsymbol{\theta}^{[k]}$ is obtained by querying the objective function. The samples $\{\boldsymbol{\theta}^{[k]}, R^{[k]}\}_{k=1...N}$ are subsequently used to compute a new search distribution. This process will run iteratively until the algorithm converges to a solution.

## 1.2 Related Work

Recent information-theoretic (IT) policy search algorithms [9] are based on the relative entropy policy search (REPS) algorithm which was proposed in [10] as a step-based policy search algorithm. However, in [9] an episode-based version of REPS that is equivalent to stochastic search was presented. The key idea behind episode-based REPS is to control the exploration-exploitation trade-off by bounding the relative entropy between the old 'data' distribution $q(\boldsymbol{\theta})$ and the newly estimated search distribution $\pi(\boldsymbol{\theta})$ by a factor $\epsilon$. Due to the relative entropy bound, the algorithm achieves a smooth and stable learning process. However, the episodic REPS algorithm uses a sample based approximation of the KL-bound, which needs a lot of samples in order to be accurate. Moreover, a typical problem of REPS is that the entropy of the search distribution decreases too quickly, resulting in premature convergence.

Taylor approximations of the KL-divergence have also been used very successfully in the area of stochastic search, resulting in natural evolutionary strategies (NES). NES uses the natural gradient to optimize the objective [2]. The natural gradient has been shown to outperform the standard gradient in many applications in machine learning [12]. The intuition of the natural gradient is that we want to obtain an update direction of the parameters of the search distribution that is most similar to the standard gradient while the KL-divergence between new and old search distributions is bounded. To obtain this update direction, a second order approximation of the KL, which is equivalent to the Fisher information matrix, is used.

Surrogate based stochastic search algorithms [6][13] have been shown to be more sample efficient than direct stochastic search methods and can also smooth out the noise of the objective function. For example, an individual optimization method is used on the surrogate that is stopped whenever the KL-divergence between the new and the old distribution exceeds a certain bound [6]. For the first time, our algorithm uses the surrogate model to compute the new search distribution analytically, which bounds the KL divergence of the new and old search distribution, in closed form.

Quadratic models have been used successfully in trust region methods for local surrogate approximation [14, 15]. These methods do not maintain a stochastic search distribution but a point estimate and a trust region around this point. They update the point estimate by optimizing the surrogate and staying in the trusted region. Subsequently, heuristics are used to increase or decrease the trusted region. In the MORE algorithm, the trusted region is defined implicitly by the KL-bound.

The Covariance Matrix Adaptation-Evolutionary Strategy (CMA-ES) is considered as the state of the art in stochastic search optimization. CMA-ES also maintains a Gaussian distribution over the problem parameter vector and uses well-defined heuristics to update the search distribution.

## 2 Model-Based Relative Entropy Stochastic Search

Similar to information theoretic policy search algorithms [9], we want to control the exploration-exploitation trade-off by bounding the relative entropy of two subsequent search distribution. However, by bounding the KL, the algorithm can adapt the mean and the variance of the algorithm. In order to maximize the objective for the immediate iteration, the shrinkage in the variance typically dominates the contribution to the KL-divergence, which often leads to a premature convergence of these algorithms. Hence, in addition to control the KL-divergence of the update, we also need to control the shrinkage of the covariance matrix. Such a control mechanism can be implemented by lower-bounding the entropy of the new distribution. In this paper, we will set the bound always to a certain percentage of the entropy of the old search distribution such that MORE converges asymptotically to a point estimate.

### 2.1 The MORE framework

Similar as in [9], we can formulate an optimization problem to obtain a new search distribution that maximizes the expected objective value while upper-bounding the KL-divergence and lower-bounding the entropy of the distribution

$$\max_{\pi} \int \pi(\boldsymbol{\theta}) \mathcal{R}_{\boldsymbol{\theta}} d\boldsymbol{\theta}, \quad \text{s.t.} \ \ \text{KL}\big(\pi(\boldsymbol{\theta})||q(\boldsymbol{\theta})\big) \leq \epsilon, \qquad H(\pi) \geq \beta, \qquad 1 = \int \pi(\boldsymbol{\theta}) d\boldsymbol{\theta}, \quad (1)$$

where $\mathcal{R}_{\boldsymbol{\theta}}$ denotes the expected objective[1] when evaluating parameter vector $\boldsymbol{\theta}$. The term $H(\pi) = -\int \pi(\boldsymbol{\theta}) \log \pi(\boldsymbol{\theta}) d\boldsymbol{\theta}$ denotes the entropy of the distribution $\pi$ and $q(\theta)$ is the old distribution. The parameters $\epsilon$ and $\beta$ are user-specified parameters to control the exploration-exploitation trade-off of the algorithm.

We can obtain a closed form solution for $\pi(\boldsymbol{\theta})$ by optimizing the Lagrangian for the optimization problem given above. This solution is given as

$$\pi(\boldsymbol{\theta}) \propto q(\boldsymbol{\theta})^{\eta/(\eta+\omega)} \exp\left(\frac{\mathcal{R}_{\boldsymbol{\theta}}}{\eta+\omega}\right), \quad (2)$$

where $\eta$ and $\omega$ are the Lagrangian multipliers. As we can see, the new distribution is now a geometric average between the old sampling distribution $q(\boldsymbol{\theta})$ and the exponential transformation of the objective function. Note that, by setting $\omega = 0$, we obtain the standard episodic REPS formulation [9]. The optimal value for $\eta$ and $\omega$ can be obtained by minimizing the dual function $g(\eta, \omega)$ such that $\eta > 0$ and $\omega > 0$, see [16]. The dual function $g(\eta, \omega)$ is given by

$$g(\eta, \omega) = \eta\epsilon - \omega\beta + (\eta + \omega) \log\left(\int q(\boldsymbol{\theta})^{\frac{\eta}{\eta+\omega}} \exp\left(\frac{\mathcal{R}_{\boldsymbol{\theta}}}{\eta+\omega}\right) d\boldsymbol{\theta}\right). \quad (3)$$

As we are dealing with continuous distributions, the entropy can also be negative. We specify $\beta$ such that the relative difference of $H(\pi)$ to a minimum exploration policy $H(\pi_0)$ is decreased for a certain percentage, i.e., we change the entropy constraint to

$$H(\pi) - H(\pi_0) \geq \gamma(H(q) - H(\pi_0)) \Rightarrow \beta = \gamma(H(q) - H(\pi_0)) + H(\pi_0).$$

Throughout all our experiments, we use the same $\gamma$ value of 0.99 and we set minimum entropy $H(\pi_0)$ of search distribution to a small enough value like $-75$. We will show that using the additional entropy bound considerably alleviates the premature convergence problem.

## 2.2 Analytic Solution of the Dual-Function and the Policy

Using a quadratic surrogate model of the objective function, we can compute the integrals in the dual function analytically, and, hence, we can satisfy the introduced bounds exactly in the MORE framework. At the same time, we take advantage of surrogate models such as a smoothed estimate in the case of noisy objective functions and a decrease in the sample complexity[2].

We will for now assume that we are given a quadratic surrogate model

$$\mathcal{R}_{\boldsymbol{\theta}} \approx \boldsymbol{\theta}^T \boldsymbol{R} \boldsymbol{\theta} + \boldsymbol{\theta}^T \boldsymbol{r} + r_0$$

of the objective function $\mathcal{R}_{\boldsymbol{\theta}}$ which we will learn from data in Section 3. Moreover, the search distribution is Gaussian, i.e., $q(\boldsymbol{\theta}) = \mathcal{N}(\boldsymbol{\theta}|\boldsymbol{b}, \boldsymbol{Q})$. In this case the integrals in the dual function given in Equation 3 can be solved in closed form. The integral inside the log-term in Equation (3) now represents an integral over an un-normalized Gaussian distribution. Hence, the integral evaluates to the inverse of the normalization factor of the corresponding Gaussian. After rearranging terms, the dual can be written as

$$g(\eta, \omega) = \eta\epsilon - \beta\omega + \frac{1}{2}\left(\boldsymbol{f}^T \boldsymbol{F} \boldsymbol{f} - \eta \boldsymbol{b}^T \boldsymbol{Q}^{-1} \boldsymbol{b} - \eta \log|2\pi \boldsymbol{Q}| + (\eta+\omega)\log|2\pi(\eta+\omega)\boldsymbol{F}|\right) \quad (4)$$

with $\boldsymbol{F} = (\eta \boldsymbol{Q}^{-1} - 2\boldsymbol{R})^{-1}$ and $\boldsymbol{f} = \eta \boldsymbol{Q}^{-1}\boldsymbol{b} + \boldsymbol{r}$. Hence, the dual function $g(\eta, \omega)$ can be efficiently evaluated by matrix inversions and matrix products. Note that, for a large enough value of $\eta$, the matrix $\boldsymbol{F}$ will be positive definite and hence invertible even if $\boldsymbol{R}$ is not. In our optimization, we always restrict the $\eta$ values such that $\boldsymbol{F}$ stays positive definite[3].

Nevertheless, we could always find the $\eta$ value with the correct KL-divergence. In contrast to MORE, Episodic REPS relies on a sample based approximation of the integrals in the dual function in Equation (3). It uses the sampled rewards $\mathcal{R}_{\boldsymbol{\theta}}$ of the parameters $\theta$ to approximate this integral.

We can also obtain the update rule for the new policy $\pi(\boldsymbol{\theta})$. From Equation (2), we know that the new policy is the geometric average of the Gaussian sampling distribution $q(\boldsymbol{\theta})$ and a squared exponential given by the exponentially transformed surrogate. After re-arranging terms and completing the square, the new policy can be written as

$$\pi(\boldsymbol{\theta}) = \mathcal{N}\left(\boldsymbol{\theta}|\boldsymbol{F}\boldsymbol{f}, \boldsymbol{F}(\eta+\omega)\right), \quad (5)$$

where $\boldsymbol{F}$, $\boldsymbol{f}$ are given in the previous section.

## 3 Learning Approximate Quadratic Models

In this section, we show how to learn a quadratic surrogate. Note that we use the quadratic surrogate in each iteration to locally approximate the objective function and not globally. As the search distribution will shrink in each iteration, the model error will also vanish asymptotically. A quadratic surrogate is also a natural choice if a Gaussian distribution is used, cause the exponent of the Gaussian is also quadratic in the parameters. Hence, even using a more complex surrogate, it can not be exploited by a Gaussian distribution. A local quadratic surrogate model provides similar second-order information as the Hessian in standard gradient updates. However, a quadratic surrogate model also has quadratically many parameters which we have to estimate from a (ideally) very small data

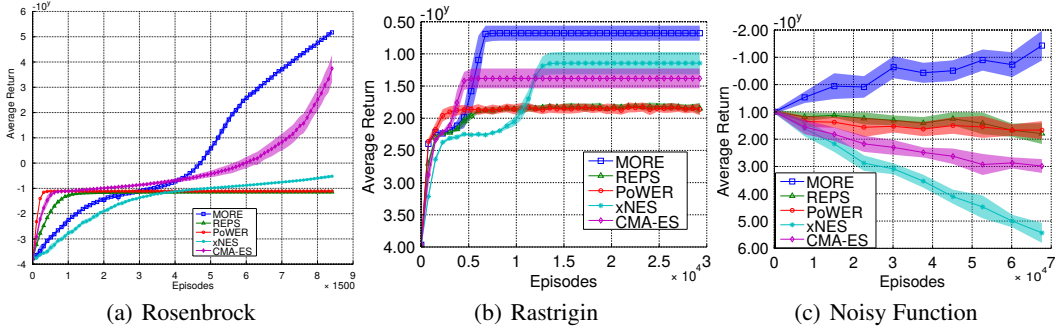

|     |     |     |
| --- | --- | --- |
| (a) Rosenbrock | (b) Rastrigin | (c) Noisy Function |

Figure 1: Comparison of stochastic search methods for optimizing the uni-modal Rosenbrock (a) and the multi modal (b) Rastrigin function. (c) Comparison for a noisy objective function. All results show that MORE clearly outperforms other methods.

set. Therefore, already learning a simple local quadratic surrogate is a challenging task. In order to learn the local quadratic surrogate, we can use linear regression to fit a function of the form $f(\boldsymbol{\theta}) = \boldsymbol{\phi}(\boldsymbol{\theta})\boldsymbol{\beta}$, where $\boldsymbol{\phi}(\boldsymbol{\theta})$ is a feature function that returns a bias term, all linear and all quadratic terms of $\boldsymbol{\theta}$. Hence, the dimensionality of $\boldsymbol{\phi}(\boldsymbol{\theta})$ is $D = 1 + d + d(d + 1)/2$, where $d$ is the dimensionality of the parameter space. To reduce the dimensionality of the regression problem, we project $\boldsymbol{\theta}$ in a lower dimensional space $l_{p \times 1} = \boldsymbol{W}\boldsymbol{\theta}$ and solve the linear regression problem in this reduced space[4]. The quadratic form of the objective function can then be computed from $\boldsymbol{\beta}$ and $\boldsymbol{W}$. Still, the question remains how to choose the projection matrix $\boldsymbol{W}$. We did not achieve good performance with standard PCA [17] as PCA is unsupervised. Yet, the $\boldsymbol{W}$ matrix is typically quite high dimensional such that it is hard to obtain the matrix by supervised learning and simultaneously avoid over-fitting. Inspired by [18], where supervised Bayesian dimensionality reduction are used for classification, we also use a supervised Bayesian approach where we integrate out the projection matrix $\boldsymbol{W}$.

### 3.1   Bayesian Dimensionality Reduction for Quadratic Functions

In order to integrate out the parameters $\boldsymbol{W}$, we use the following probabilistic dimensionality reduction model

$$p(r_*|\boldsymbol{\theta}_*, \boldsymbol{D}) = \int p(r_*|\boldsymbol{\theta}_*, \boldsymbol{W})p(\boldsymbol{W}|\boldsymbol{D})d\boldsymbol{W}, \tag{6}$$

where $r_*$ is prediction of the objective at query point $\boldsymbol{\theta}_*$, $\boldsymbol{D}$ is the training data set consisting of parameters $\boldsymbol{\theta}^{[k]}$ and their objective evaluations $R^{[k]}$. The posterior for $\boldsymbol{W}$ is given by Bayes rule, i.e., $p(\boldsymbol{W}|\boldsymbol{D}) = p(\boldsymbol{D}|\boldsymbol{W})p(\boldsymbol{W})/p(\boldsymbol{D})$. The likelihood function $p(\boldsymbol{D}|\boldsymbol{W})$ is given by

$$p(\boldsymbol{D}|\boldsymbol{W}) = \int p(\boldsymbol{D}|\boldsymbol{W}, \boldsymbol{\beta})p(\boldsymbol{\beta})d\boldsymbol{\beta}, \tag{7}$$

where $p(\boldsymbol{D}|\boldsymbol{W}, \boldsymbol{\beta})$ is the likelihood of the linear model $\beta$ and $p(\boldsymbol{\beta})$ its prior. For the likelihood of the linear model we use a multiplicative noise model, i.e., the higher the absolute value of the objective, the higher the variance. The intuition behind this choice is that we are mainly interested in minimizing the relative error instead of the absolute error[5]. Our likelihood and prior is therefore given by

$$p(\boldsymbol{D}|\boldsymbol{W}, \boldsymbol{\beta}) = \prod_{k=1}^{N} \mathcal{N}(R^{[k]}|\boldsymbol{\phi}(\boldsymbol{W}\boldsymbol{\theta}^{[k]})\boldsymbol{\beta}, \sigma^2|R^{[k]}|), \quad p(\boldsymbol{\beta}) = \mathcal{N}(\boldsymbol{\beta}|\boldsymbol{0}, \tau^2\boldsymbol{I}), \tag{8}$$

Equation 7 is a weighted Bayesian linear regression model in $\boldsymbol{\beta}$ where the weight of each sample is scaled by the absolute value of $|R^{[k]}|^{-1}$. Therefore, $p(\boldsymbol{D}|\boldsymbol{W})$ can be obtained efficiently in closed form. However, due to the feature transformation, the output $R^{[k]}$ depends non-linearly on the projection $\boldsymbol{W}$. Therefore, the posterior $p(\boldsymbol{W}|\boldsymbol{D})$ cannot be obtained in closed form any more. We use a simple sample-based approach in order to approximate the posterior $p(\boldsymbol{W}|\boldsymbol{D})$. We use $K$ samples from the prior $p(\boldsymbol{W})$ to approximate the integrals in Equation (6) and in $p(\boldsymbol{D})$. In this case, the predictive model is given by

$$p(r_*|\boldsymbol{\theta}_*, \boldsymbol{D}) \approx \frac{1}{K} \sum_i p(r_*|\boldsymbol{\theta}_*, \boldsymbol{W}_i) \frac{p(\boldsymbol{D}|\boldsymbol{W}_i)}{p(\boldsymbol{D})}, \tag{9}$$

where $p(\boldsymbol{D}) \approx 1/K \sum_i p(\boldsymbol{D}|\boldsymbol{W}_i)$. The prediction for a single $\boldsymbol{W}_i$ can again be obtained by a standard Bayesian linear regression. Our algorithm is only interested in the expectation $\mathcal{R}_{\boldsymbol{\theta}} = \mathbb{E}[r|\boldsymbol{\theta}]$ in the form of a quadratic model. Given a certain $\boldsymbol{W}_i$, we can obtain a single quadratic model from $\phi(\boldsymbol{W}_i\boldsymbol{\theta})\boldsymbol{\mu}_\beta$, where $\boldsymbol{\mu}_\beta$ is the mean of the posterior distribution $p(\boldsymbol{\beta}|\boldsymbol{W}, \boldsymbol{D})$ obtained by Bayesian linear regression. The expected quadratic model is then obtained by a weighted average over all $K$ quadratic models with weight $p(\boldsymbol{D}|\boldsymbol{W}_i)/p(\boldsymbol{D})$. Note that with a higher number of projection matrix samples(K), the better the posterior can be approximated. Generating these samples is typically inexpensive as it just requires computation time but no evaluation of the objective function. We also investigated using more sophisticated sampling techniques such as elliptical slice sampling [19] which achieved a similar performance but considerably increased computation time. Further optimization of the sampling technique is part of future work.

## 4 Experiments

We compare MORE with state of the art methods in stochastic search and policy search such as CMA-ES [1], NES [2], PoWER [20] and episodic REPS [9]. In our first experiments, we use standard optimization test functions [21], such as the the Rosenbrock (uni modal) and the Rastrigin (multi modal) functions. We use a 15 dimensional version of these functions.

Furthermore, we use a 5-link planar robot that has to reach a given point in task space as a toy task for the comparisons. The resulting policy has 25 parameters, but we also test the algorithms in high-dimensional parameter spaces by scaling the robot up to 30 links (150 parameters). We subsequently made the task more difficult by introducing hard obstacles, which results in a discontinuous objective function. We denote this task hole-reaching task. Finally, we evaluate our algorithm on a physical simulation of a robot playing beer pong. The used parameters of the algorithms and a detailed evaluation of the parameters of MORE can be found in the supplement.

### 4.1 Standard Optimization Test Functions

We chose one uni-modal functions $f(\boldsymbol{x}) = \sum_{i=1}^{n-1}[100(x_{i+1} - x_i^2)^2 + (1 - x_i)^2]$, also known as Rosenbrock function and a multi-modal function which is known as the Rastgirin function $f(x) = 10n + \sum_{i=1}^{n}[x_i^2 - 10\cos(2\pi x_i)]$. All these functions have a global minimum equal $f(\boldsymbol{x}) = 0$. In our experiments, the mean of the initial distributions has been chosen randomly.

**Algorithmic Comparison.** We compared our algorithm against CMA-ES, NES, PoWER and REPS. In each iteration, we generated 15 new samples [6]. For MORE, REPS and PoWER, we always keep the last $L = 150$ samples, while for NES and CMA-ES only the 15 current samples are kept[7]. As we can see in the Figure 1, MORE outperforms all the other methods in terms of learning speed and final performance in all test functions. However, in terms of the computation time, MORE was 5 times slower than the other algorithms. Yet, MORE was sufficiently fast as one policy update took less than $1s$.

**Performance on a Noisy Function.** We also conducted an experiment on optimizing the Sphere function where we add multiplicative noise to the reward samples, i.e., $y = f(\boldsymbol{x}) + \epsilon|f(\boldsymbol{x})|$, where $\epsilon \sim \mathcal{N}(0, 1.0)$ and $f(\boldsymbol{x}) = \boldsymbol{x}\boldsymbol{M}\boldsymbol{x}$ with a randomly chosen $\boldsymbol{M}$ matrix.

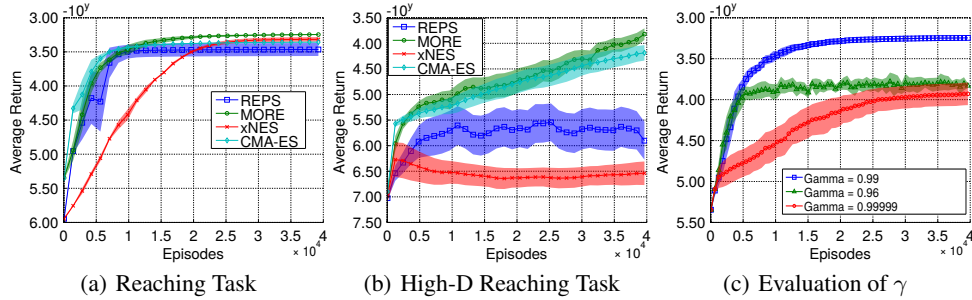

Figure 2: (a) Algorithmic comparison for a planar task (5 joints, 25 parameters). MORE outperforms all the other methods considerably.(b) Algorithmic comparison for a high-dimensional task (30 joints, 150 parameters). The performance of NES degraded while MORE could still outperform CMA-ES. (c) Evaluation of the entropy bound $\gamma$. For a low $\gamma$, the entropy bound is not active and the algorithm converges prematurely. If $\gamma$ is close to one, the entropy is reduced too slowly and convergence takes long.

Figure 1(c) shows that MORE successfully smooths out the noise and converges, while other methods diverge. The result shows that MORE can learn highly noisy reward functions.

## 4.2 Planar Reaching and Hole Reaching

We used a 5-link planar robot with DMPs [22] as the underlying control policy. Each link had a length of $1m$. The robot is modeled as a decoupled linear dynamical system. The end-effector of the robot has to reach a via-point $\boldsymbol{v}_{50} = [1, 1]$ at time step $50$ and at the final time step $T = 100$ the point $\boldsymbol{v}_{100} = [5, 0]$ with its end effector. The reward was given by a quadratic cost term for the two via-points as well as quadratic costs for high accelerations. Note that this objective function is highly non-quadratic in the parameters as the via-points are defined in end effector space. We used 5 basis functions per degree of freedom for the DMPs while the goal attractor for reaching the final state was assumed to be known. Hence, our parameter vector had 25 dimensions. The setup, including the learned policy is shown in the supplement.

**Algorithmic Comparison.** We generated 40 new samples. For MORE, REPS, we always keep the last $L = 200$ samples, while for NES and CMA-ES only the 40 current samples are kept. We empirically optimized the open parameters of the algorithms by manually testing 50 parameter sets for each algorithm. The results shown in Figure 2(a) clearly show that MORE outperforms all other methods in terms of speed and the final performance.

**Entropy Bound.** We also evaluated the entropy bound in Figure 2(c). We can see that the entropy constraint is a crucial component of the algorithm to avoid the premature convergence.

**High-Dimensional Parameter Spaces.** We also evaluated the same task with a 30-link planar robot, resulting in a 150 dimensional parameter space. We compared MORE, CMA, REPS and NES. While NES considerably degraded in performance, CMA and MORE performed well, where MORE found considerably better policies (average reward of -6571 versus -15460 of CMA-ES), see Figure 2(b). The setup with the learned policy from MORE is depicted in the supplement.

We use the same robot setup as in the planar reaching task for hole reaching task. For completing the hole reaching task, the robot's end effector has to reach the bottom of a hole (35cm wide and $1$ m deep) centering at $[2, 0]$ without any collision with the ground or the walls, see Figure 3(c). The reward was given by a quadratic cost term for the desired final point, quadratic costs for high accelerations and additional punishment for collisions with the walls. Note that this objective function is discontinuous due to the costs for collisions. The goal attractor of the DMP for reaching the final state in this task is unknown and is also learned. Hence, our parameter vector had 30 dimensions.

**Algorithmic Comparison.** We used the same learning parameters as for the planar reaching task. The results shown in Figure 3(a) show that MORE clearly outperforms all other methods. In this task, NES could not find any reasonable solution while Power, REPS and CMA-ES could only learn sub-optimal solutions. MORE could also achieve the same learning speed as REPS and CMA-ES, but would then also converge to a sub-optimal solution.

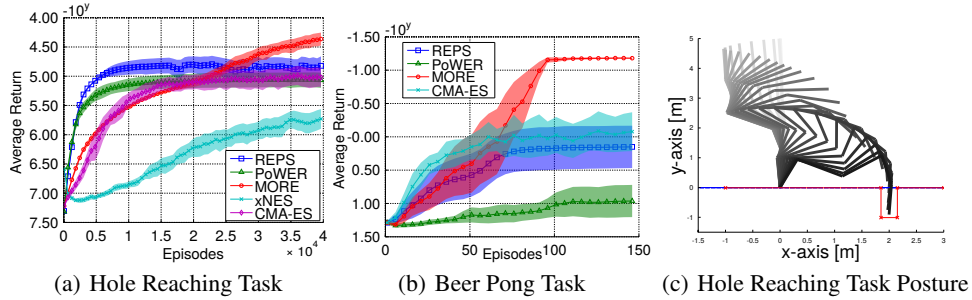

(a) Hole Reaching Task      (b) Beer Pong Task      (c) Hole Reaching Task Posture

Figure 3: (a) Algorithmic comparison for the hole reaching task. MORE could find policies of much higher quality. (b) Algorithmic comparison for the beer pong task. Only MORE could reliably learn high-quality policies while for the other methods, even if some trials found good solutions, other trials got stuck prematurely.

## 4.3 Beer Pong

In this task, a seven DoF simulated barrett WaM robot arm had to play beer-pong, i.e., it had to throw a ball such that it bounces once on the table and falls into a cup. The ball was placed in a container mounted on the end-effector. The ball could leave the container by a strong deceleration of the robot's end-effector. We again used a DMP as underlying policy representation, where we used the shape parameters (five per DoF) and the goal attractor (one per DoF) as parameters. The mean of our search distribution was initialized with imitation learning. The cup was placed at a distance of 2.2m from the robot and it had a height of 7cm. As reward function, we computed the point of the ball trajectory after the bounce on the table, where the ball is passing the plane of the entry of the cup. The reward was set to be 20 times the negative squared

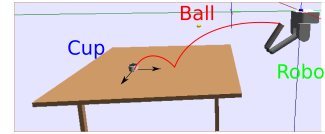

(a) Beer Pong Task

Figure 4: The Beer Pong Task. The robot has to throw a ball such that it bounces of the table and ends up in the cup.

distance of that point to the center of the cup while punishing the acceleration of the joints. We evaluated MORE, CMA, PoWER and REPS on this task. The setup is shown in Figure 4 and the learning curve is shown in Figure 3(b). MORE was able to accurately hit the ball into the cup while the other algorithms couldn't find a robust policy.

## 5 Conclusion

Using KL-bounds to limit the update of the search distribution is a wide-spread idea in the stochastic search community but typically requires approximations. In this paper, we presented a new model-based stochastic search algorithm that computes the KL-bound analytically. By relying on a Gaussian search distribution and on locally learned quadratic models of the objective function, we can obtain a closed form of the information theoretic policy update. We also introduced an additional entropy term in the formulation that is needed to avoid premature shrinkage of the variance of the search distribution. Our algorithm considerably outperforms competing methods in all the considered scenarios. The main disadvantage of MORE is the number of parameters. However based on our experiments, these parameters are not problem specific.

## Acknowledgment

This project has received funding from the European Unions Horizon 2020 research and innovation programme under grant agreement No #645582 (RoMaNS) and the first author is supported by FCT under grant SFRH/BD/81155/2011.

## Footnotes

[1]Note that we are typically not able to obtain the expected reward but only a noisy estimate of the underlying reward distribution.

[2]The regression performed for learning the quadratic surrogate model estimates the expectation of the objective function from the observed samples.

[3]To optimize $g$, any constrained nonlinear optimization method can be used[13].

[4]$\boldsymbol{W}_{(p \times d)}$ is a projection matrix that projects a vector from a $d$ dimension manifold to a $p$ dimension manifold.

[5]We observed empirically that such relative error performs better if we have non-smooth objective functions with a large difference in the objective values. For example, an error of 10 has a huge influence for an objective value of $-1$, while for a value of $-10000$, such an error is negligible.

[6] We use the heuristics introduced in [1, 2] for CMA-ES and NES

[7] NES and CMA-ES algorithms typically only use the new samples and discard the old samples. We also tried keeping old samples or getting more new samples which decreased the performance considerably.

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
