[Supplementary Material]

# Supplementary

**Abbas Abdolmaleki**[1,2,3]**, Rudolf Lioutikov**[4]**, Nuno Lau**[1]**, Luis Paulo Reis**[2,3]**,
Jan Peters**[4,6]**, and Gerhard Neumann**[5]

1: IEETA, University of Aveiro, Aveiro, Portugal
2: DSI, University of Minho, Braga, Portugal
3: LIACC, University of Porto, Porto, Portugal
4: IAS, 5: CLAS, TU Darmstadt, Darmstadt, Germany
6: Max Planck Institute for Intelligent Systems, Stuttgart, Germany
{Lioutikov,peters,neumann}@ias.tu-darmstadt.de
{abbas.a, nunolau}@ua.pt, lpreis@dsi.uminho.pt

## 1   Algorithm Robustness Analysis

The open parameter parameters of the MORE algorithm are as follows:

| $\tau^2$ | $\sigma^2$ | $p$ | $K$ | $\epsilon$ | $\gamma$ |
|---|---|---|---|---|---|
| **100** | **100** | **5** | **1000** | **1** | **0.99** |

In order to show the robustness of our method with respect to algorithm parameters, we evaluate the performance of our algorithm with different parameter settings $\sigma^2$, $\tau^2$ and $K$ of the model learning approach on optimising Rosenbrock function. The Figure 1 shows that the algorithm is able to perform well with a wide range of parameters.

We also compared the Bayesian dimensionality reduction technique introduced in the paper with using a standard PCA for dimensionality reduction. We evaluated both techniques for a different number of latent space dimensions. The results in Figure 2(a) show the average reward over the first 300 iterations of planar reaching task with 25 parameters. As we can see, Bayesian dimensionality reduction performed consistently better than PCA and was also more robust to the chosen number of latent space dimensions.

we further evaluated the KL divergence parameter $\epsilon$ of MORE for three different values while other parameters of the algorithm are constant. Figure 2(b) shows that with different $\epsilon$ values we get different results. With a big $\epsilon$ the algorithm will exploit the learned surrogate more and might be misled by a false optimum while with too small $\epsilon$ the algorithm will have slow convergence rate.

## 2   Learned Policies

**High Dimensional Plannar Reaching Task** figure 2(c) shows a 30-link robot that has to reach a via-point $\boldsymbol{v}_{50} = [1, 1]$. The via-point is indicated by the red cross. The postures of the resulting motion are shown as overlay, where darker postures indicate a posture which is close in time to the via-point. The results show the policy learned by MORE. The algorithmic comparison has been given in the paper.

**Hole Reaching Task** Figure 3 shows a 5-link robot that has to reach the bottom of a hole centred at $[2, 0]$ while avoiding the collision with the ground (red line). The postures of the resulting motion learned by CMA-ES, REPS and MORE are shown as overlay, where darker postures indicate a posture which is close in time to the hole. The results show that while MORE learns a good and robust policy, other algorithms suffer from premature convergence.

(a) Different $\tau^2$

(b) Different $\sigma^2$

(c) Different $K$

Figure 1: The performance of MORE for different model learning parameters. In this experiment we keep all parameters constant and only evaluate one parameter in each plot. The results show that MORE robustly outperforms the best of CMA with all different parameters set but one.

(a) Projection Methods

(b) KL Divergence Bound $\epsilon$

(c) MORE High Dimensional Planar Reaching Task

Figure 2: (a) Comparison of supervised Bayesian dimensionality reduction and PCA. The plots show the average performance value for the first 300 iterations of the planar reaching task with 25 parameters. Our Bayesian dimensionality reduction method outperformed PCA for all number of dimensions of the latent space. The best performance could be achieved with five projections. (b) We evaluated the parameter $\epsilon$ of MORE on planar reaching task with 25 parameters, While we kept other parameters constant, The results show that too small $\epsilon$ results in slow convergence while too big $\epsilon$ will mislead the algorithm by false optimum introduced by the learned surrogate model, the best result is achieved with $\epsilon$ 1. (c) A 30-link robot has to reach a via-point $\boldsymbol{v}_{50} = [1, 1]$. The via-point is indicated by the red cross. The postures of the resulting motion are shown as overlay, where darker postures indicate a posture which is close in time to the via-point. The results show the policy learned by MORE

(a) CMA-ES Hole Reaching Task Policy

(b) REPS Hole Reaching Task Policy

(c) MORE Hole Reaching Task Policy

Figure 3: A 5-link robot has to reach a bottom of a hole centred at $[2, 0]$ while avoiding the collision with the ground (red line). The postures of the resulting motion are shown as overlay, where darker postures indicate a posture which is close in time to the hole. The results show that while MORE learns a good and robust policy, other algorithms suffer from premature convergence.