[Reviews · NeurIPS 2015]

Submitted by Assigned_Reviewer_1

Comments: - The writing quality is currently well below the standard of accepted NIPS papers, and needs to be improved. - I think the paper could be improved by comparing to a sample based approximation of Equation 3, along the lines of Episodic REPS, but with the entropy bound added. It's hard to tell otherwise where the main performance improvement is coming from, i.e. the bound or the quadratic approximation. - In the definition of H(pi), what is s? - The crude sample based approximation in Equation 9 is not very convincing and quite simplistic. I wouldn't be confident about this generalizing to other domains. - It would help to describe explicitly how the best setup was found for each algorithm in the experimental comparison. - In section 4.2 (Entropy bound), I think you mean Figure 4(c). - I think it would be worthwhile citing and maybe comparing your work to "CLOP: Confident local optimization for noisy black box parameter tuning", LNCS, 2012, which also leverages local quadratic approximation.
Summary: Promising experimental results, but to me it is unclear what is causing the main difference in performance. The writing quality could also be improved.

Submitted by Assigned_Reviewer_2

Summary: The paper introduces an algorithm (MORE) for black box optimization that constructs local quadratic ``surrogate'' models based on recent observations.

By using a (simpler than the true function) quadratic model, the iterative step of refining the reward parameters can be computed in closed form (though the models themselves seem to be built with a form of sampling). This approach allows for a more sample efficient and robust search procedure, which is shown to outperform state-of-the-art methods in terms of samples and converged parameters on a number of simple functions as well as some very complex robotics tasks.

Review: The new algorithm performs much better than the state-of-the-art algorithms in a wide range of experiments and is applicable in a very important problem setting.

I appreciate the wide range of problems that the authors used and I think they make a very strong case for the new algorithm.

I do think there is some room for improvement in what experimental conditions were considered, how some of the results are presented, and in the general description of the algorithm's properties, which I cover here:

First and foremost, the experiments section does not contain any results on computational speed and in fact, computation time is not mentioned until the next-to-last sentence in the paper.

But there, it is mentioned that MORE takes more time than all the other approaches (though we are told it is fast enough, taking less than a second).

It would be better to see some actual comparisons of computation time from the experiments (perhaps in place of one of the robotic arm diagrams), and the computation time issue ought to be mentioned much earlier rather than buried at the very end of the paper.

I do not understand why experiments were not conducted using the exact solution of the quadratic modeling problem described in the paragraph starting on line 196.

I do understand this would require many more samples than the dimensionality-reduction approach, but it seems possible the exact solution might converge to a better set of parameters.

Why not show this approach on some of the graphs (at least in the simple problems) and also report how many more samples it needs than reduced-dimension approach?

Or, if the number of samples is always prohibitive, state that definitively to mark the exact solution as intractable.

There is a disconnect between the description of the approach in the early sections, which focus on the ability to do closed form calculations, and the computation of the posterior (line 249) of the quadratic models, which requires sampling.

I understand that the change in parameters can now be calculated in closed form, which is a nice achievement, but it seems like the burden of sampling may have just been pushed deeper into the algorithm (in the posterior calculation).

If this is the case, it should be mentioned earlier on that the algorithm still requires sampling at some level.

A basic premise of the paper (line 106) is that keeping the previous and next reward functions close to one another ``controls'' the exploration-exploitation dilemma.

But there are certainly cases (say when the current function creates terrible behavior) where the proper thing to do is to completely change the function parameters.

A better description around line 106 of when this algorithm is allowed to make such large leaps (or why it does not) would make the approach more understandable.

At line 101, it is mentioned that CMA-ES is ``well known as an ad-hoc algorithm'' but no citation is given and no real evidence of that fact is presented.

Graphs 4b and 4c seem to be referenced incorrectly in the text (b is referenced as the entropy graph, but its really c, and the reverse for the high dimensional graph)

Typos: Line 21 - frail to converge prematurely - converge prematurely. Line 55 -allow to - allow us to Line 107 - search distribution - search distributions Line 110 - control - controlling Line 111 - Such control - Such a control Line 154 - from using - of using Line 155 - decreased the - a decrease in the
Summary: The new algorithm performs much better than the state-of-the-art algorithms in a wide range of experiments and is applicable in a very important problem setting.

There were a number of points in the text where the rationale for experimental decisions need to be clarified (described in review).

Submitted by Assigned_Reviewer_3

I should start by saying I'm probably not the best person to review this paper, as it is somewhat outside my main areas of expertise.

That said, the paper is relatively clear and both the problem (black box optimization) and the suggested solution (model-based relative entropy stochastic search) are straightforward to understand.

The method looks promising.

However, I miss a comparison to other methods that map a quadratic model to find the appropriate direction of search, e.g., "The BOBYQA algorithm for bound constrained optimization without derivatives" by Powell (2009).

I'm not sure about the validity of the claim that "For the first time, our algorithm uses the surrogate model to compute the new search distribution analytically", and it seems a comparison is in order (although the precise algorithms are clearly different, so the authors can still claim some novelty).

Hopefully the authors can clarify.
Summary: The paper present a new black-box optimization algorithm.

The results are promising, but there is lots of room for further understanding, in particular about how to best tune the algorithm and if it can be sped up.

Submitted by Assigned_Reviewer_4

This paper adds a local quadratic model of the objective function to REPS, a popular and apparently very good policy search algorithm. This allows the algorithm to run fewer actual objective function evaluations (which correspond to actual robot time in the robotics case). Some theory is included which bounds the relative entropy of the policy jump, and so prevents moving too far (as the quadratic approximation could be bad).

At a high level, this is a good idea and the empirical results seem strong. It would be interesting to evaluate how much of an effect the avoiding-premature-convergence properties of the algorithm have vs. the fewer-samples parts. It looks like the new algorithm gets a better quality solution in almost every benchmark problem, but not a sample-complexity reduction in all of them (though i performs reasonably well in most).

I'd discourage the use of the term "fitness function". How about "objective function"? Fitness is specific to the GA community, whereas the algorithm you're talking about here has a wider audience.

If I understand line 113 correctly, the authors are enforcing a geometrically shrinking entropy. Why is that reasonable?

Bayesian dimensionality reduction seems overkill. Have you really exhausted all simpler methods? L1 regularization?

Generally the paper is very sloppily written. Some small writing things:

o "policy search of robot motor skills" - just say "policy search".

o "of them are frail to converge prematurely" - "may converge prematurely"

o "These algorithms have weak assumption on the type of fitness function."

- "These algorithms make only weak assumptions ..."

o "surrogate allows to compute" - allows US to compute

o "avoids over fitting makes the" - "avoids over-fitting COMMA making the algorithm"

o "to high dimensional the problems" delete "the"

o "[10]" is not a noun. This problem occurs throughout the paper and is aggressively irritating.

o "strategies(NES)" - add a space.

o "optimize the performance" - delete "the".

o "two subsequent search distribution" - distributionS

o "addition to control the" - in addition to controlling the

o "Similar as in" - "Similar to"

o No spaces before footnotes. Footnotes go immediately after a period.

o Spaces after sub-figure labels (e.g., "(b)Rastrigin").

o Line 240 ends suddenly.

o Footnote 4 ... I don't even know where to start. That is a sentence. It starts with a capital letter. It ends with a period.

etc. etc.
Summary: A good paper with a nice idea, badly written.

Author Feedback
Author rebuttal: We thank all the reviewers. And we apologize for typos, grammar mistakes, unclear notations and missing citations. They will be corrected such that the overall writing meet NIPS standards. The below clarifications will be added in the paper or supplement either in form of texts or figures.

#Review 1

1- Optimizing the Rosenbrock function, "REPS with entropy bound" outperforms REPS with a final return of -7.14 compared to -11.60 which is worse than -1.24e-04 achieved by MORE. The reason is that, maximum likelihood estimate in REPS uncontrollably reduces the entropy. The entropy bound reduces this effect, but it can't fully counteract it. Also, in figure 4(c), MORE with a loose entropy bound suffers from premature convergence. Therefore, the entropy bound and "satisfying the bounds analytically" are both critical for the performance of MORE.

2- s in H(pi) is a typo.

3- In Eq.9, the more number of projection matrices (K), the better the precision of the posterior will be. Generating these matrices is typically inexpensive as it requires computation time but no evaluation of the objective function.

4- We optimized the open parameters of the algorithms by manually testing 50 parameter sets for each algorithm.

#Review 2

1- Computation time mainly depends on the population size and number and dimension of projection matrices. In standard function experiments, it took less than 1s per iteration and for higher dim. problems it goes up to 3s.

2- Using the surrogate without dim. reduction works nicely in presence of enough samples of current distribution (> 0.5*(n+1)*(n+2)), which is infeasible for high dim. problems and expensive objective functions.

3- About Eq.9 please see answer #3 for review #1.

4- We don't necessarily keep the reward functions close together but the distributions. The distribution should never have a leap or jump as we want to perform local search and such jumps would cause instabilities.

5- In the paper "Wierstra, et al. Natural evolution strategies. 2008", it mentions, CMA-ES has an ad-hoc nature. But following the review #6 we will remove it.

#Review 3

1- We performed several experiments to test the sample complexity. In general, MORE is relatively insensitive to the number of samples taken per iteration provided that the epsilon and the gamma are adapted with the number of samples. The less samples we use, the less we should allow to move the distribution. We will add such an experiment.

2- With the shrinking variance, MORE converges asymptotically to a point estimate. It also makes sense to set the entropy dynamically.

3- We tried L1 regularization but couldn't achieve good results. Also, the computation time of L1 limited us from performing a large number of experiments. Please see answer #2 for review #2.

#Review 4

1- We will compare against NEWUOA and BOBYQA, but still, these methods unlike MORE don't maintain a stochastic distribution but a point estimate and a trust region around this point (which is arguably similar to a spherical Gaussian). However, they update the point estimate with gradient descent on the surrogate and subsequently, using heuristics, they increase or decrease the trusted region. Therefore, they can't do the update analytically. MORE uses surrogate to compute the mean and full covariance of the Gaussian in one step by analytically solving the opt. problem in line 122.

2- We will extend our discussion on parameter tuning and computational demands. (Please see supplement and reviews #2,#3)

#Review 5

In principle, our algorithm can be applied to any stochastic search problem, as every function can be approximated by a quadratic function locally. The presented contributions, i.e. the additional entropy bound and the analytic solution of the optimization problem are crucial to make the REPS competitive to other stochastic search algorithms.

Please note that we use the quadratic model to locally approximate the objective function and not globally and as the search distribution will shrink in each iteration, the model error will vanish asymptotically. Also, quadratic surrogate is a natural choice if a Gaussian distribution is used, because the exponent of the Gaussian is also quadratic in the parameters. Hence, a more complex surrogate couldn't be exploited by a Gaussian distribution. Also, our experiments showed that MORE works favorably for non-quadratic objective functions.
Also, quadratic models have been used successfully, in trust region methods for local approximation. See, e.g. UOBYQA (Powell, 2002).

#Review 6

To optimize g, any constrained nonlinear optimization method can be used [13]. We use fmincon in matlab.
H(pi0) is the minimum entropy of the distribution. We set it to a small value like -75.
In our experiments, the means of the initial distributions have been chosen randomly. Initializations will be clarified. Please also see the answers #4 and #5 for reviews #1 and #2.